# Gut Microbiota and Metabolites in Atrial Fibrillation Patients and Their Changes after Catheter Ablation

Kang Huang,[a] Yuegang Wang,[b] Yang Bai,[a] Qiuyan Luo,[a] Xuancai Lin,[a] Qiuyu Yang,[a] Shihao Wang,[a] Hongjie Xin[a]

aDepartment of Gastroenterology, Nanfang Hospital, Southern Medical University, Guangzhou, China
bDepartment of Cardiology, State Key Laboratory of Organ Failure Research, Nanfang Hospital, Southern Medical University, Guangzhou, China

**ABSTRACT** The gut microbiota has been shown to be associated with multiple cardiovascular diseases, but there is little research on the gut microbiota and atrial fibrillation (AF); thus, how the gut microbiota and metabolites change in AF patients after catheter ablation is unclear. In this study, we used 16S rRNA high-throughput sequencing and nontargeted metabolomic detection to conduct horizontal and longitudinal analyses of the gut microbiota and metabolites of AF patients. Compared with a control group, species richness and diversity increased significantly in AF patients. Among them, opportunistic pathogenic bacteria, such as *Klebsiella*, *Haemophilus*, *Streptococcus*, and *Enterococcus*, were significantly increased, and symbiotic bacteria, such as *Agathobacter* and *Butyrivibrio*, were significantly reduced. After catheter ablation, intestinal symbiotic bacteria (*Lactobacillus*, *Agathobacter*, *Lachnospira*, etc.) were increased in most AF patients, while pathogenic bacteria (*Ruminococcus*, etc.) were reduced. Moreover, in AF patients, caffeine, which was negatively correlated with *Klebsiella*, was downregulated, and estradiol and ascorbic acid, which were positively correlated with *Agathobacter*, were also downregulated. After catheter ablation, citrulline, which was positively correlated with *Ralstonia* and *Lactobacillus*, was increased. Oleanolic acid, which was negatively correlated with *Ralstonia* was downregulated. In conclusion, our results not only show overall changes in the gut microbiota and metabolites in AF patients but also indicate their changes in the short term after catheter ablation. These data will provide novel possibilities for the future clinical diagnosis and treatment of AF.

**IMPORTANCE** Gut microbiota and metabolites play a very important role in human health and can not only assess human health but also treat and prevent diseases. We analyzed the characteristics of the microbiota and metabolites in the human gut and found the effect of disease on gut microbiota and metabolites, which may be of important value in the pathogenesis of atrial fibrillation. At the same time, we also observed dynamic changes in gut microbiota and metabolites with the intervention of catheter ablation, which was not available in previous studies.

**KEYWORDS** gut microbiota, gut metabolites, atrial fibrillation

Atrial fibrillation (AF) is a supraventricular tachyarrhythmia with uncoordinated atrial activation and, consequently, ineffective atrial contraction; it is one of the most common and severe arrhythmias worldwide, with a lifetime risk of 26% for men and 23% for women over 40 years of age (1). AF can lead to decreased atrial function and an abnormal ventricular rate, which can significantly reduce cardiac output and cause a series of symptoms, such as palpitation, syncope, dyspnea, heart failure, and cardiac arrest (1). Moreover, AF even leads to cerebral apoplexy caused by left atrial mural thrombosis, which is an important cause of AF disability and death (2). More attention should be given to the prevention and early diagnosis of AF, which is helpful not only for reducing the risks of stroke, heart failure, and other related complications but also for delaying the

Address correspondence to Yuegang Wang, 1248508@qq.com, or Yang Bai, 13925001665@163.com.

The authors declare no conflict of interest.

progression of AF-induced atrial electrical and structural remodeling through early intervention. At present, the diagnosis of AF mainly relies on electrocardiography (ECG) (3); however, AF cannot always be detected in a single ECG examination, and many patients do not have obvious symptoms in the early stage (4, 5), which also presents a challenge in early diagnosis of AF. Although catheter ablation is effective in the treatment of AF, it is an invasive treatment and has disadvantages, such as complexity, high cost, surgery-related complications, and a risk of postoperative recurrence, rendering it unacceptable to many patients. Therefore, further study of the pathological mechanism of AF and identification of new diagnosis and treatment strategies is important to compensate for the deficiencies of existing diagnosis and treatment methods.

As the largest and most abundant bacterial reservoir in the body, the gut can affect heart function both directly (via metabolites) and indirectly (via the immune system) (6). Inflammation, as a potential pathogenesis mechanism of AF, can increase susceptibility to AF by changing the atrial electrophysiology and structural substrates and is related to heterogeneous atrial conduction and AF triggers. Overgrowth of Gram-negative gut bacteria may lead to the accumulation of lipopolysaccharides (LPS), which can cause a series of inflammatory responses, such as activation of NLRP3 inflammasomes, which leads to activation of caspase-1, secretion of interleukin-1$\beta$ (IL-1$\beta$) and IL-18, changes in intestinal permeability, and promotion of endotoxin transport and inflammatory cytokine accumulation, creating a vicious cycle. LPS can also combine with Toll-like receptor 4 (TLR4) to activate NF-$\kappa$B signal transduction, which can lead to lipid accumulation and induce overexpression of IL-6, IL-8, monocyte chemotactic proteins, and various cell adhesion molecules, causing vascular inflammation and participating in myocardial cell apoptosis, hypertrophy, fibrosis, and other processes (7, 8). At present, there have been few studies on the gut microbiota and metabolites involved in AF, which have produced inconsistent results. Zuo et al. (9) found that individuals with AF showed increased gut microbiota richness and diversity, but Tabata et al. (10) found that the richness of gut microbiota in AF patients was decreased. Papandreou et al. (11, 12) found that plasma trimethylamine N-oxide (TMAO) levels were generally not elevated in AF and were not associated with AF progression, but Svingen et al. (13, 14) reported that plasma TMAO levels were associated with AF. Therefore, more evidence is needed to clarify the correlation of gut microbiota and metabolites with AF. In addition, the gut microbiota and metabolome are not fixed, especially in a population in a disease state. When surgical intervention occurs, the host state changes, leading to changes in the gut microbiota and metabolome. Whether the gut microbiota and metabolic patterns of AF patients change after catheter ablation is unknown at present.

Therefore, more emphasis should be placed on early diagnosis and treatment of AF, especially asymptomatic AF, and the management of complications. In addition, we aimed to identify new pathophysiological mechanisms, explore potential biomarkers, and develop noninvasive treatment strategies. Here, for the first time, we conducted a cross-sectional and longitudinal study of gut microbiota and metabolites in AF patients, aiming to provide new evidence for the relationship of AF with gut microbiota and metabolites and to dynamically evaluate the treatment and prognosis of catheter ablation from the perspective of gut microbiota and metabolites. This study helps explore the pathophysiological mechanism of AF and provides an important basis for early diagnosis and treatment of AF and the management of its complications.

## RESULTS

**Baseline characteristics of the study cohort.** The clinical characteristics of all participants are presented in Table 1. There was no significant difference between patients with AF and controls in terms of uric acid, creatinine, total bilirubin, or alanine aminotransferase (ALT). Moreover, we incidentally found that the total cholesterol, triglyceride, and low-density lipoprotein (LDL) levels in AF patients were lower than those in the control group, but those of the vast majority of subjects were within the normal range. These results may be associated with the dose and frequency of lipid-lowering

**TABLE 1** Baseline characteristics of the study cohort

| Characteristic[a] | AF[b] | Control[b] | P value |
|---|---|---|---|
| No. | 36 | 30 | /[c] |
| Age/yrs | 56.00 (47.50, 63.75) | 50.50 (40.00, 57.00) | 0.123 |
| Male/female sex | 25/11 | 22/8 | 0.728 |
| BMI | 24.11 ± 3.16 | 25.25 ± 3.03 | 0.135 |
| HTN | 19 | 15 | 0.822 |
| T2DM | 4 | 3 | 1.00 |
| Total cholesterol | 4.08 ± 0.95 | 4.92 ± 0.83 | <0.001 |
| Triglyceride | 1.38 (1.01, 1.67) | 1.53 (1.24, 3.16) | 0.017 |
| LDL | 2.60 (2.10, 3.14) | 3.12 (2.59, 3.70) | 0.011 |
| Creatinine | 84.50 (68.50, 95.00) | 75.00 (67.50, 86.00) | 0.209 |
| Uric acid | 383.81 ± 18.26 | 388.06 ± 22.53 | 0.303 |
| Total bilirubin | 11.05 (8.53, 16.05) | 12.00 (7.73, 15.30) | 0.949 |
| ALT | 17.00 (12.00, 23.75) | 17.00 (14.00, 21.50) | 0.877 |
| | | | |
| Drug use | | | |
| ARB/ACEI | 13 | 10 | 0.814 |
| $\beta$-Receptor blockers | 20 | 8 | 0.265 |
| $Ca^{2+}$ channel blocker | 5 | 9 | 0.111 |
| Diuretic | 4 | 0 | 0.172 |
| Statins | 15 | 7 | 0.116 |
| Ezetimibe | 2 | 5 | 0.290 |
| Aspirin | 2 | 6 | 0.158 |
| Metformin | 2 | 1 | 1.00 |
| Amiodarone | 4 | 0 | 0.172 |
| Oral anticoagulation therapy | 19 | 0 | <0.001 |
| Proton pump inhibitors | 11 | 3 | 0.042 |

[a]HTN, hypertension; T2DM, type 2 diabetes; LDL, low-density lipoprotein; ALT, alanine aminotransferase; ARB, angiotensin receptor antagonist; ACEI, angiotensin converting enzyme inhibitor.
[b]Data are presented as mean ± standard deviation or median (interquartile range), as appropriate.
[c]/, no data.

drugs because most AF patients receive lipid reduction treatment during hospitalization. In terms of drug use, there were significant differences between the two groups in oral anticoagulant therapy and proton pump inhibitors (PPIs). Although we collected drug use information for each subject, the specific dose and frequency could not be accurately analyzed, especially for nonhospitalized patients.

**Alteration of the gut microbiota in patients with AF.** In total, the AF and control groups included 2,563 operational taxonomic units (OTUs) and shared 1,189 OTUs. Among them, 897 specific OTUs were found only in patients with AF, and 477 were found only in controls (Fig. 1A). We found that the Shannon index and Chao1 index values in the AF group were higher than those in the control group (Shannon index: AF group 5.03 versus control group 4.78, $P = 0.035$; Fig. 1B; Chao1 index: AF group 416.59 versus control group 387.52, $P = 0.027$; Fig. 1C), which showed that the gut microbial richness and diversity in the AF group were much higher. The *Firmicutes*/*Bacteroidetes* ratio was not significantly different between the two groups (AF group 1.25 versus control group 1.15, $P = 0.90$). Sixty-six samples were divided into 4 clusters by principal-coordinate analysis (PCoA) based on the Jensen-Shannon divergence (JSD; Fig. 1D and E). Although only two samples were clustered as enterotype 1, we still kept them to reflect the authenticity of the clinical sample. Therefore, AF patients may trend more toward enterotypes characterized by *Bacteroides* and *Prevotella*, and enterotypes characterized by *Bacteroides* accounted for the largest proportion in both the AF and control groups (Fig. 1F).

At the phylum level, the main bacteria in the two groups were *Firmicutes*, *Bacteroidota*, *Proteobacteria*, *Actinobacteria*, and *Fusobacteriota* (AF group: *Firmicutes* 49.27%, *Bacteroidota* 33.80%, *Proteobacteria* 10.70%, *Actinobacteria* 2.25%, and *Fusobacteriota* 2.67%; control group: *Firmicutes* 50.76%, *Bacteroidota* 34.45%, *Proteobacteria* 9.16%, *Actinobacteria* 2.59%, and *Fusobacteriota* 1.57%; Fig. 2A). The results at the genus level are displayed in Fig. 2B (AF group: *Bacteroides* 26.15%, *Blautia* 10.20%, *Faecalibacterium* 6.45%, *Prevotella* 3.98%,

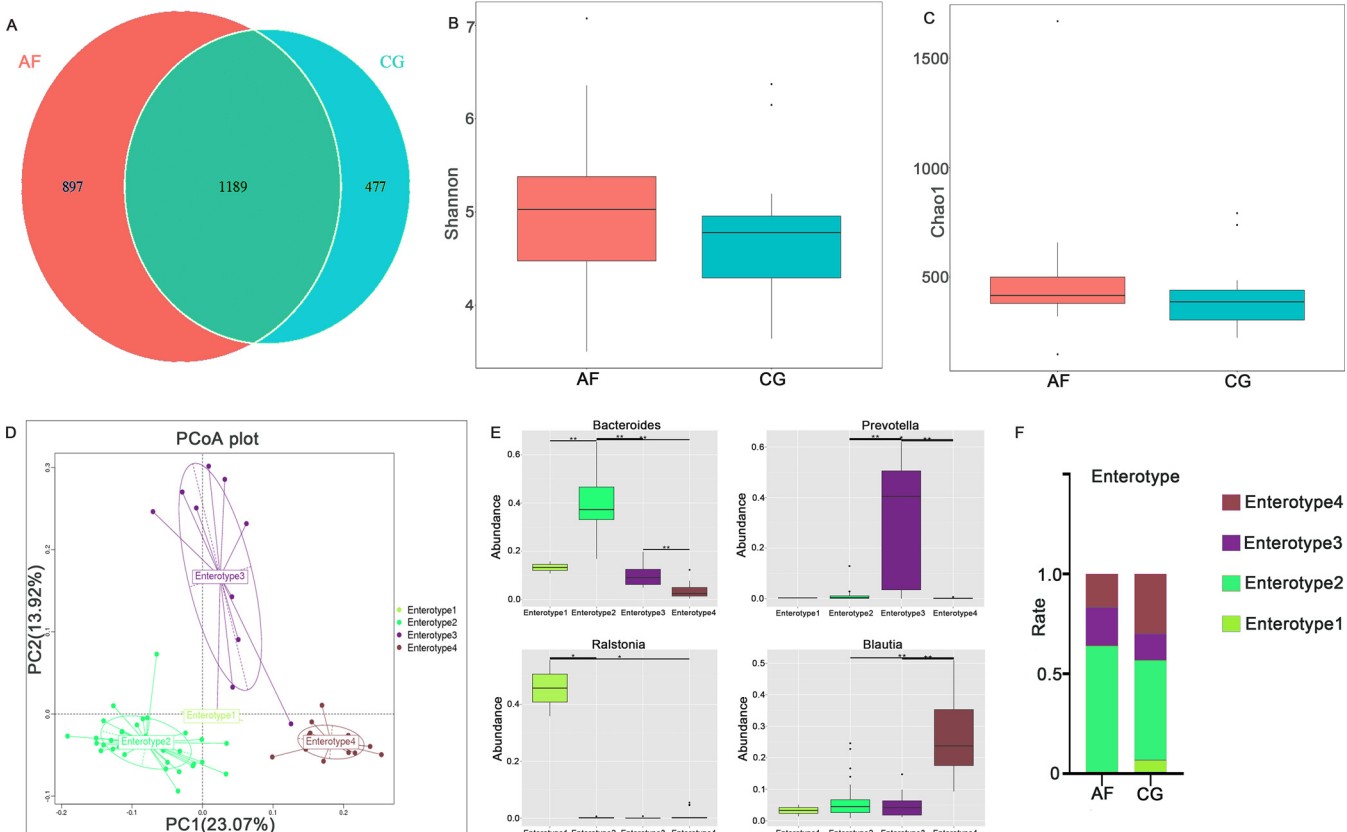

**FIG 1** Gut microbiota diversity and enterotypes in patients with AF. (A) Venn diagram based on OTUs. (B and C) Alpha-diversity index (Shannon index [B]; Chao1 index [C]) box-type diagram of the two groups. Boxes represent the interquartile ranges, and lines inside the boxes denote medians. (D) Sixty-six samples were clustered into 4 enterotypes by PCoA of JSD values at the genus level. (E) Relative abundances of the top genera in each enterotype (*Ralstonia* in enterotype 1, *Bacteroides* in enterotype 2, *Prevotella* in enterotype 3, and *Blautia* in enterotype 4). Boxes represent the interquartile ranges, and lines inside the boxes denote medians. (F) Proportion of each enterotype in the two groups; AF, atrial fibrillation; CG, control group.

and *Klebsiella* 3.85%; control group: *Bacteroides* 24.57%, *Blautia* 10.62%, *Prevotella* 7.69%, *Faecalibacterium* 4.48%, and *Agathobacter* 4.10%), and the bar plots for all individuals are displayed in Fig. C and D in the supplemental material. Moreover, at the phylum level, there were no significant differences in the main phyla (top 10) ($P \geq 0.05$). At the genus level, the relative abundances of 71 species were significantly changed ($P < 0.05$), and the first 24 different species are shown in Fig. 2C. *Klebsiella*, *Megamonas*, *Methylobacterium-Methylorubrum*, *Parabacteroides*, *Streptococcus*, *Weissella*, *Alistipes*, *Haemophilus*, and *Enterococcus*, etc. were significantly more abundant in the AF group than in the control group, whereas *Agathobacter*, *Butyrivibrio*, and *Eubacterium ventriosum* were remarkably less abundant than in the control group. Linear discriminant analysis effect size (LEfSe) analysis (with the linear discriminant analysis [LDA] score set to 3) showed the species differences between the two groups at various classification levels Fig. A in the supplemental material. In the AF group, *Haemophilus*, *Alistipes*, *Enterococcus*, *Weissella*, *Parabacteroides*, *Megamonas*, *Streptococcus*, and *Klebsiella* were obviously enriched, while in the control group, *Agathobacter* was enriched. The enrichment results of *Agathobacter*, *Haemophilus*, *Alistipes*, *Enterococcus*, *Weissella*, *Parabacteroides*, *Megamonas*, *Streptococcus*, and *Klebsiella* were consistent with the results of the Wilcoxon rank sum test between groups, indicating the large contributions of these species in both the AF and control populations.

Considering that the influence of drugs on gut microbiota cannot be ignored, we divided the AF patients into groups according to whether they were treated with PPIs or anticoagulant therapy (dabigatran or rivaroxaban) before specimen retention. We found that the AF patients treated with PPIs had higher *Streptococcus*, *Faecalibacterium*, *Weissella*, and *Lactobacillus* abundances (Fig. 2D). Moreover, when treated with anticoagulants,

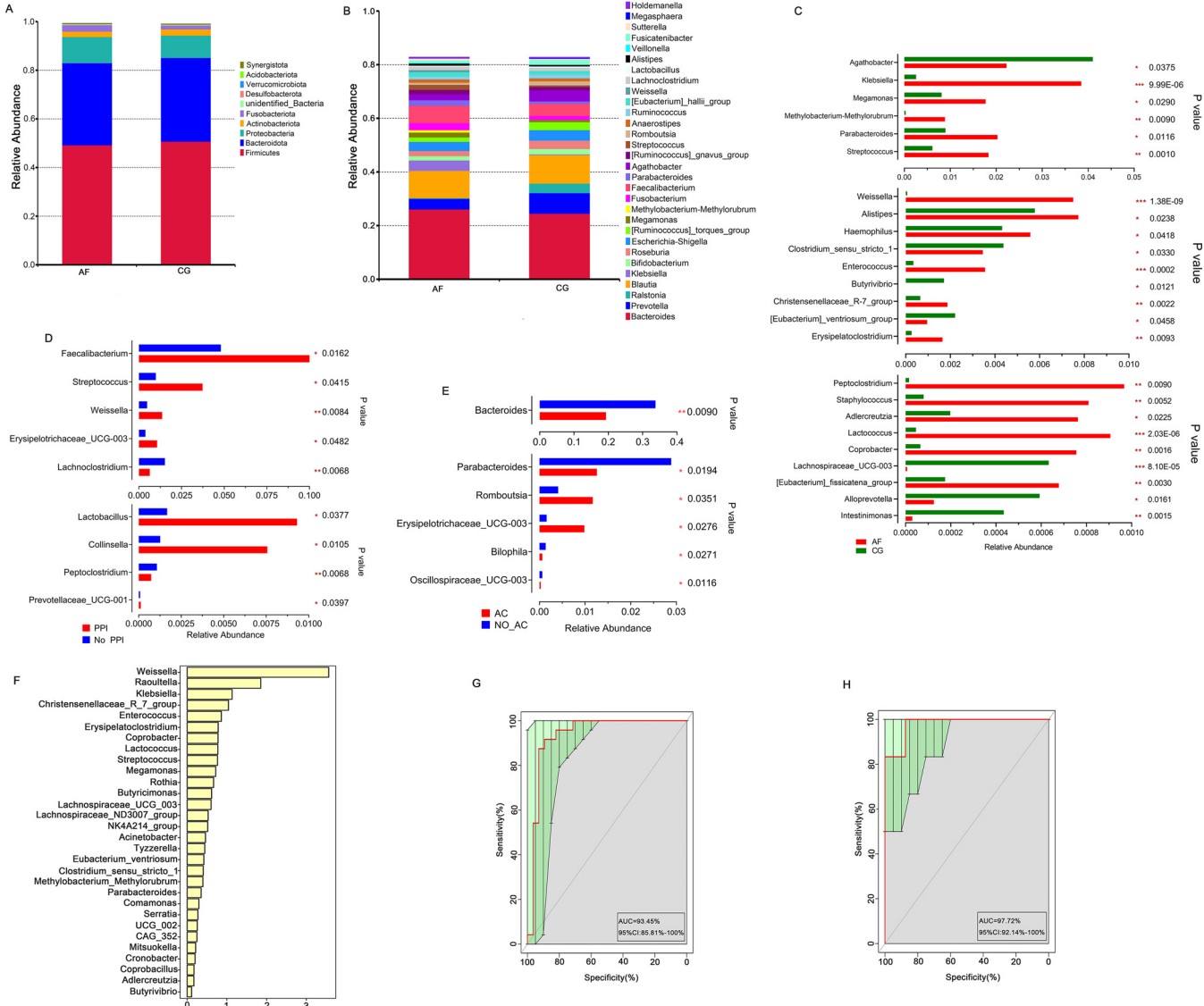

**FIG 2** Differences in gut microbiota between AF patients and control subjects. (A and B), Histograms of relative species abundance (top 10 at the phylum level [A] and top 30 at the genus level [B]). (C) Top 24 species with significant differences in relative abundance at the genus level ($P < 0.05$, Wilcoxon rank sum test). (D) Bacteria with significant differences between AF patients with and without PPI use. (E) Bacteria with significant differences between AF patients with and without anticoagulation treatment. (F) Ranking of contributions to AF obtained by random forest analysis of the first 30 species of bacteria. (G and H) Receiver operating characteristic (ROC) curves of the training set (G) and test set (H). The AUCs were 93.45% (G) and 97.92% (H). The green area represents the 95% CI (85.81% to 100% [G]; 92.14% to 100% [H]); *, $P < 0.05$; **, $P < 0.01$; ***, $P < 0.001$; PPI, proton pump inhibitor; AC, anticoagulation; AF, atrial fibrillation.

*Bacteroidetes* and *Parabacteroides* were less abundant, but *Romboutsia* and *Erysipelotrichaceae_ UCG-003* were more abundant (Fig. 2E).

Then, random forest analysis based on the relative abundance of species at the genus level showed that the discriminant model based on 30 species with significant differences could effectively separate the experimental group from the control group, and of those species, *Weissella*, *Raoultella*, *Klebsiella*, *Christensenellaceae_R-7_group*, and *Enterococcus* contributed the most (Fig. 2F). In the training set ($n = 52$), the area under the receiver operating characteristic curve (AUC) was 93.45% (95% confidence interval [CI] of 85.81% to 100%; Fig. 2G), indicating that the AF patients could be effectively distinguished from the control population. In the test set ($n = 14$), the AUC for distinguishing the AF patients from the control group was 97.92% (95% CI of 92.14% to 100%; Fig. 2H). Therefore, the model based on gut microbiota could separate the AF patients from the control population, suggesting that gut microbiota can serve as a potential biomarker for predicting and identifying AF patients.

**Changes in gut microbiota in AF patients after catheter ablation.** We also conducted short-term postoperative follow-ups of the AF patients after catheter ablation and collected stool samples, with follow-ups including patients with AF prior to catheter ablation (AF1 group), 2 days after catheter ablation (average 2.2 days; AF2 group) and 1 month after catheter ablation (average 35.4 days; AF3 group). We found that there were 2,692 OTUs in the three groups combined, and 1,026 OTUs were shared. The AF1 group had 280 unique OTUs, the AF2 group had 503 unique OTUs, and the AF3 group had 199 OTUs (Fig. 3A). First, we found that there were no significant differences in changes in the Shannon index and Chao1 index values in the AF patients at 2 days and 1 month after surgery (Shannon index: 5.07 versus 4.90 [$P = 0.525$] and 5.09 versus 4.90 [$P = 0.483$]; Chao1 index: 448.07 versus 405.59 [$P = 0.217$] and 397.02 versus 405.59 [$P = 0.728$]; Fig. 3B and C), although we found that in postoperative patients, Shannon index and Chao1 index values increased more than they decreased (Fig. 3D and E). Then, we also calculated the Bray-Curtis distance between the microbiota in each sample after surgery and before surgery and found that at 2 days after operation and 1 month after operation, there was no significant difference in the Bray-Curtis distance ($P = 0.102$, paired Wilcoxon signed-rank test; Fig. 3F). PCA based on OTUs showed that the postoperative microbial community changes became more obvious with longer follow-up times (Fig. 3G). Enterotype analysis showed that all samples were divided into 3 enterotypes (Fig. 3H and I). Only one sample clustered into enterotype 1, which was characterized by *Ralstonia* and came from the AF2 group. Enterotype 2, which was characterized by *Bacteroides*, accounted for the largest proportion in the three groups. Compared to that in the AF1 group, the distribution of enterotypes in the AF2 and AF3 groups did not change significantly ($P \geq 0.05$, Fisher's exact test; Fig. 3J), indicating that the enterotypes of the AF patients did not change significantly in the short term after catheter ablation.

At the phylum level, *Firmicutes*, *Bacteroidota*, and *Proteobacteria* had the highest abundances in all three groups (Fig. 3K; the bar plots for all individuals are displayed in Fig. E in the supplemental material), and the abundance of each phylum (top 5) reached the highest or lowest values 1 month after surgery (Fig. 3M). Among the phyla, the relative abundance of *Bacteroidota* was increased 2 days after surgery and 1 month after surgery, but the paired Wilcoxon signed-rank test (AF2 versus AF1 and AF3 versus AF1) did not show significant differences ($P \geq 0.05$). The ratio of *Firmicutes*/*Bacteroides* in the two pairs of compatibility groups showed no significant differences (AF2 versus AF1, $P = 0.078$; AF3 versus AF1, $P = 0.306$).

Then, we paired and compared the top 40 bacteria (Fig. 3L; the bar plots for all individuals are displayed in Fig. F in the supplemental material) at the genus level and screened out species with significant differences ($P < 0.05$). Comparing the preoperative and 2-day postoperative abundances, a total of 9 bacterial genera underwent significant changes; 6 bacterial genera increased in abundance, including *Prevotella*, *Lactobacillus*, *Sutterella*, *Alistipes*, *Parabacteroides*, and *Streptococcus*, and 3 bacterial genera showed reduced abundance, including *Blautia*, *Ruminococcus_gnavus_group*, and *Anaerostipes* (Fig. 3N). Eleven bacterial genera underwent significant changes in the first month after surgery, with 8 bacterial genera increasing in abundance, including *Agathobacter*, *Subdoligranulum*, *Lachnospira*, *Prevotella*, *Lactobacillus*, *Streptococcus*, *Haemophilus*, and *Coprococcus*, and 3 bacterial genera decreasing in abundance, including *Ralstonia*, *Weissella*, and *Dialister* (Fig. 3O), among which *Prevotella*, *Lactobacillus*, and *Streptococcus* were increased at both 2 days and 1 month after surgery. LEfSe analysis of the three groups (with the LDA score set to 4) showed the species differences and developmental distribution at each classification level. *Clostridia* played an important role before surgery; *Streptococcus*, *Lactobacillaceae*, *Lactobacillus*, and *Ralstonia* played important roles 2 days after surgery, and *Prevotella*, *Prevotellaceae*, and *Agathobacter* played important roles 1 month after surgery (Fig. B in the supplemental material).

**Changes in gut metabolomics before and after catheter ablation in AF patients.** To study the changes in the gut metabolome of AF patients, we conducted a nontargeted metabolomics study by liquid chromatography-mass spectrometry (LC-MS) of the stools of some subjects. First, we divided the three groups of samples into two pairs for comparison (AF1 versus CG and AF1 versus AF2; the AF1 group represented

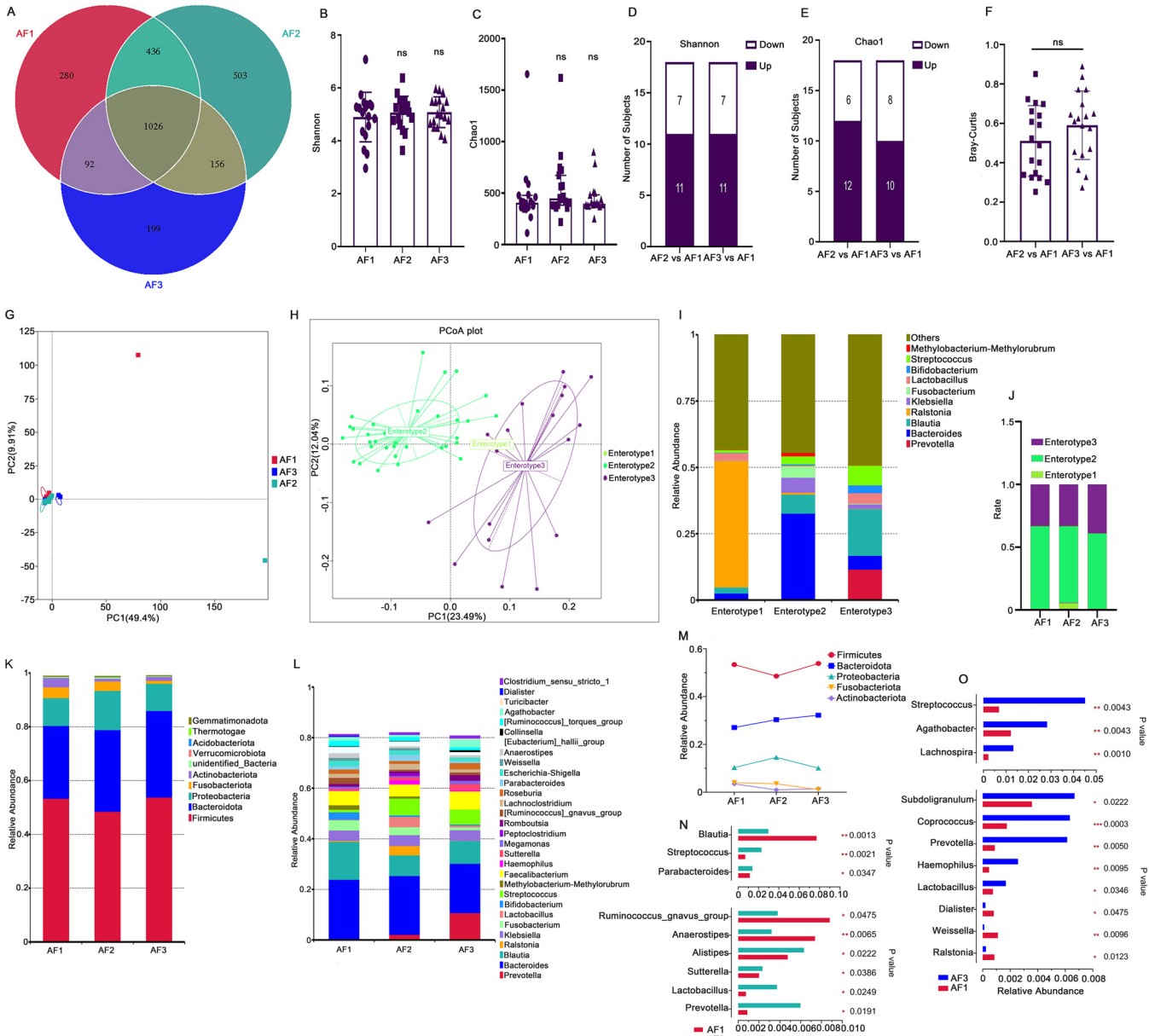

**FIG 3** Shifts in gut microbiota in AF patients after catheter ablation. (A) Venn diagram based on OTUs showing the number of unique and shared OTUs among the three groups. (B and C) Alpha-diversity index values at different time points after operation, as determined using the paired sample Wilcoxon signed-rank test (ns, $P \geq 0.05$). (D and E) Number of AF patients with increases and decreases in each group after operation compared with before the operation. (F) Bray-Curtis distances between the microbiota at each time point after surgery and before surgery for each AF patient (significance was measured using the Wilcoxon rank-sum test; ns, $P \geq 0.05$). (G) Three groups from PCA based on OTUs. The circle shows a CI of 95%. (H) By PCoA of the JSD values at the genus level, 54 samples were clustered into three enterotypes. (I) Bar graph of the relative abundances of bacteria that make up each enterotype. Enterotype 1 was characterized by *Ralstonia*, enterotype 2 was characterized by *Bacteroides*, and enterotype 3 was characterized by *Blautia*. (J) Distribution of each enterotype in the three groups. (K and L) Histograms of the relative abundance of species (top 10 at the phylum level [K] and top 30 at the genus level [L]). (M) Relative abundance in the three groups at the phylum level. (N and O), At the genus level, the relative abundance of species with significant differences. Paired analysis was performed on AF patients for preoperative and postoperative comparisons (paired sample Wilcoxon signed-rank test, $P < 0.05$).

AF patients before catheter ablation, the AF2 group represented AF patients 2 days after catheter ablation, and the CG group represented control subjects) and then generated a PCA three-dimensional (3D) chart and partial least-squares discrimination analysis (PLS-DA) scatter chart to compare the overall metabolic differences between the two groups. In the positive and negative ion modes, each compared pair showed clear separation (Fig. 4A to H), indicating that overall metabolism was significantly different between the AF patients and the control population and that the overall metabolism

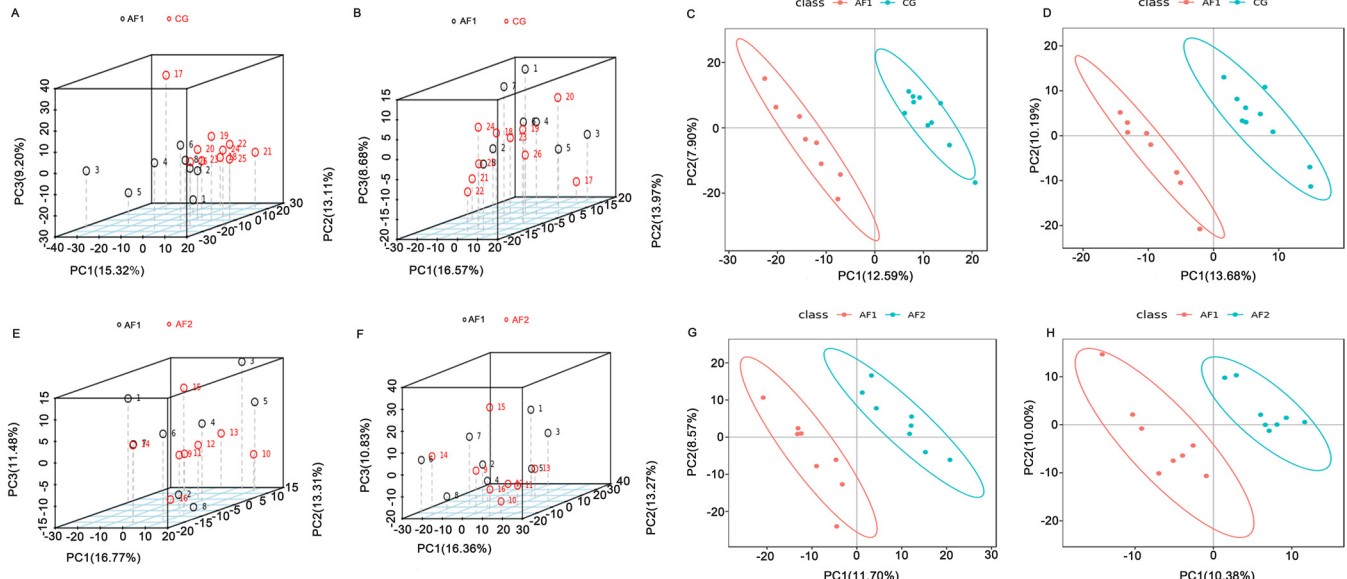

**FIG 4** Changes in metabolic patterns in AF patients. (A and B) Positive ion (ES⁺; A) and negative ion (ES⁻; B) mode PCA 3D images of AF patients and the control group. (C and D) ES⁺ (C) and ES⁻ (D) mode PCA 3D images of AF patients before and after catheter ablation. (E and F) ES⁺ (E) and ES⁻ (F) mode PLS-DA of AF patients and the control group. (G and H) ES⁺ (G) and ES⁻ (H) mode PLS-DA of AF patients before and after catheter ablation.

of the AF patients before and after catheter ablation treatment was also significantly different.

Next, we identified and screened differential metabolites, with the screening threshold set to variable importance in the projection (VIP) of >1.0, fold change (FC) of >1.5, or FC of <0.667 with a $P$ value of <0.05. In our study, compared with those in the control group, 224 metabolites were decreased, and 46 metabolites were increased in the AF patients, and 2 days after catheter ablation, 74 metabolites were decreased, and 28 metabolites were increased in the AF patients (Tables S1). Notably, we excluded subjects with diabetes and tried to match the subjects for other basic characteristics, including age, sex, body mass index (BMI), history of hypertension (HTN), and drug use (Table S2). We found that flavin adenine dinucleotide (FAD), riboflavin-5-phosphate, inosine, dehydroepiandrosterone (DHEA), estradiol, caffeine, salicylic acid, ascorbic acid, eicosapentaenoic acid (EPA), and oleanolic acid were significantly downregulated in AF patients compared with the normal control group, while *N*-acetylmethionine and 3-hydroxy-3-methylbutanoic acid were significantly upregulated (Fig. 5A). To explore the relationships between differential metabolites and gut microbiota imbalance, we performed a correlation analysis between 10 species of bacteria and the main differential metabolites with significant differences between the two groups (the results are shown in Fig. 5B). Caffeine and its metabolites (paraxanthine and theobromine) were negatively correlated with *Megamonas*, *Klebsiella*, and *Methylorubrum*, while *Agathobacter* was positively correlated with guanosine, estradiol, ascorbic acid, and *N*-acetylmethionine. In AF patients, in the short term after catheter ablation treatment, citrulline, 6-hydroxymelatonin, and homovanillic acid were significantly upregulated, while pelargonidin, α-linolenic acid, linolelaidic acid, oleanolic acid, and phosphatidylcholine (PC) were significantly downregulated (Fig. 5C). Correlation analysis between the different metabolites before and after catheter ablation treatment and the different species (Fig. 5D) showed that pelargonidin was negatively correlated with *Haemophilus*, *Ralstonia*, and *Lactobacillus* and positively correlated with *Anaerostipes* and *Ruminococcus_gnavus_group*. Citrulline was positively correlated with *Ralstonia*, *Lactobacillus*, and *Alistipes*, while oleanolic acid was negatively correlated with *Ralstonia*.

## DISCUSSION

In this study, we conducted horizontal and longitudinal analyses of the gut microbiota and metabolites of AF patients, providing new evidence for the relationship of

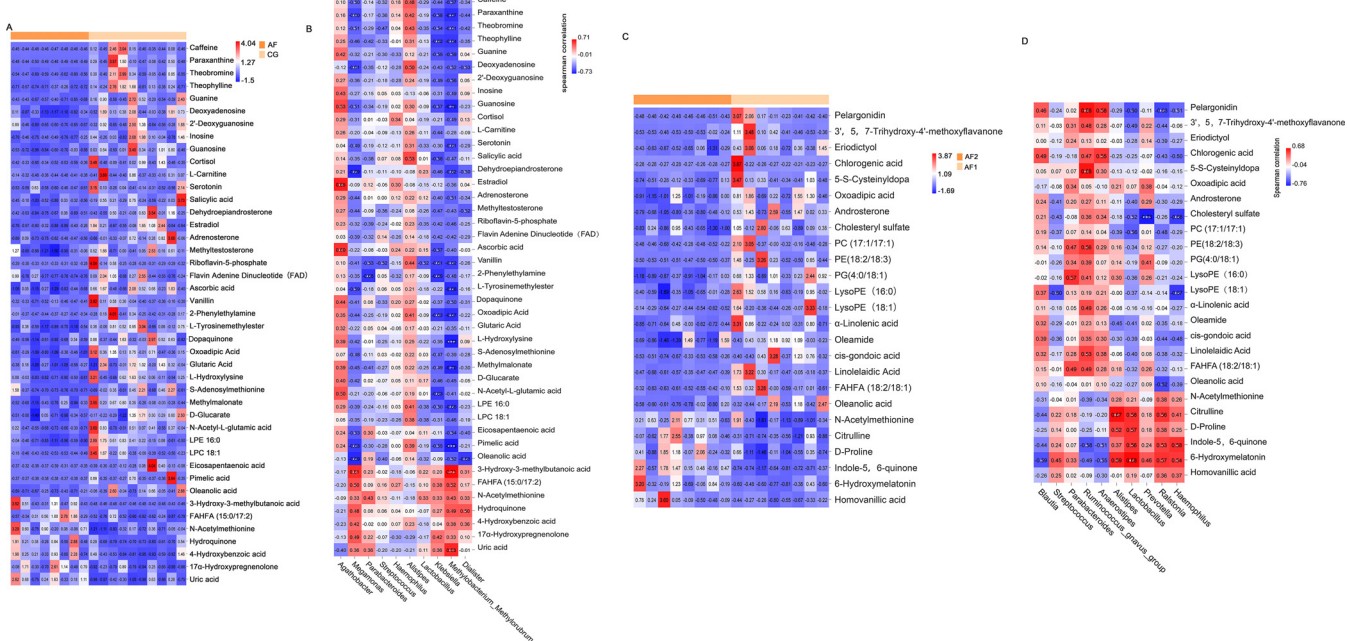

**FIG 5** Correlation analysis between the major differential metabolites and gut microbiota. (A) Expression heat map of 43 major compounds altered in AF patients compared with the control group. The relative quantitative values of different compounds were normalized and converted. Different colors represent different expression intensities; high expression intensity is shown in red, and low expression intensity is shown in blue. (B) Correlation analysis between the main metabolites and 10 main species with significant differences between the AF and control groups. (C) Expression heat map of 25 main compounds altered in AF patients after surgery compared with before surgery. (D) Correlation analysis between the major metabolites and 10 different species with significant differences before and after AF catheter ablation. In the correlation analysis (Spearman correlations), blue indicates a negative correlation, red indicates a positive correlation, and $0.01 < P < 0.05$ is marked as *, $0.001 < P < 0.01$ is marked as **, $P \leq 0.001$ is marked as ***, and $P \geq 0.05$ is not marked.

AF with gut microbiota and metabolites. We also found changes in gut microbiota and metabolites in AF patients in the short term after catheter ablation treatment.

To the best of our ability, we matched confounders such as age, sex, and BMI, which was consistent with a previous study in China (9), and the richness and diversity of gut microbiota in the AF patients remained higher than those in the control group. However, this result contradicted a study by Japanese scholars (10), which may be greatly related to the difference in geographical location and eating habits and should be confirmed with multicenter data with a larger sample size. Species richness and diversity mainly reflect the overall distribution of species. Currently, many studies have shown that species richness and diversity increase in a variety of cardiocerebrovascular diseases, such as stable angina pectoris (15) and ST-segment elevation myocardial infarction (STEMI) (16). In stroke patients, the richness, diversity, and phylogenetic diversity of gut microbiota increased significantly, possibly due to overgrowth of multiple pathogenic bacteria or opportunistic pathogens and decreases in symbiotic and beneficial bacteria (17), which is also consistent with the characteristics of the gut microbiota of AF patients. In addition, our results showed that species richness and diversity in the AF patients did not change significantly in the short term after catheter ablation. Thus, increased species diversity may be one of the features of AF and may not be significantly altered by catheter ablation in the short term.

In the current study, there were fewer enterotypes characterized by *Blautia* in AF. *Blautia* has been found to have the potential to maintain or improve disease status associated with metabolic syndrome (18). Moreover, the distribution of enterotypes in AF did not change significantly in the short term after catheter ablation.

Our study showed that *Klebsiella*, *Haemophilus*, *Streptococcus*, *Weissella*, *Alistipes*, and *Enterococcus* were significantly increased in patients with AF, while *Agathobacter* and *Butyrivibrio* were decreased. Overgrowth of Gram-negative pathogenic bacteria may promote the development of AF through the LPS inflammatory response system.

In addition, *Klebsiella*, *Streptococcus*, and *Enterococcus* have been shown to be correlated with the production of trimethylamine (TMA). The elevation of plasma TMAO levels can predict thrombosis in AF patients, and TMAO can be used as an independent predictor of ischemic stroke (19, 20). *Agathobacter* and *Butyrivibrio* are associated with the production of short-chain fatty acids (such as butyric acid) and lactic acid (21), and short-chain fatty acids play an indispensable role in maintaining the integrity of the intestinal barrier and have beneficial cardiovascular regulatory effects (22). In contrast, our study showed that in the short term after catheter ablation, intestinal symbiotic and beneficial bacteria were increased in most AF patients (*Lactobacillus*, *Prevotella*, *Agathobacter*, *Coprococcus*, *Lachnospira*, etc.), while pathogenic bacteria and opportunistic pathogens (*Ruminococcus*, *Ralstonia*, *Weissella*, etc.) were reduced. The results indicated that catheter ablation is helpful for the recovery and stabilization of gut microbiota.

Incidentally, we found that *Streptococcus* and *Haemophilus* were also increased after catheter ablation, which might be associated with the increased use of some drugs, such as PPIs. We grouped the AF patients by whether PPIs were used and found that patients who used PPIs showed a greater abundance of *Streptococcus*. Moreover, previous studies have shown that PPIs can increase the species abundance of *Streptococcus*, *Staphylococcus*, *Enterococcus*, and potentially pathogenic *Escherichia coli*, which increases the risk of intestinal infection (23). Nevertheless, the majority of AF patients increased the use of PPIs after catheter ablation to prevent gastrointestinal ulcers.

Many metabolites have been shown to influence cardiovascular health. In our study, downregulation of flavin mononucleotide (FMN) and FAD in AF may be related to disordered riboflavin metabolism, which can lead to mitochondrial dysfunction by reducing energy metabolism levels (24) and may affect the normal function of the heart and increase susceptibility to AF. In addition, FAD can significantly increase the expression of short-chain acyl coenzyme A dehydrogenase and inhibit pathomyocardial hypertrophy and fibrosis (25). Inosine can induce the improvement of endothelium-dependent vasodilation, exhibit antiplatelet properties, activate endothelial nitric oxide synthase (eNOS), and reduce p38MAPK/NF-$\kappa$B pathway expression in aortic tissue (26). DHEA and estradiol, which participate in steroid hormone biosynthesis, have been confirmed to affect the cardiovascular system through regulation. DHEA can act directly on target tissues, such as endothelial cells, smooth muscle cells, and cardiac muscle cells, showing a cardioprotective effect (27). Estradiol metabolites may exert a protective effect on cardiac vessels by inhibiting vascular smooth muscle cells and cardiac fibroblasts and improving the function of vascular endothelial cells (28). In addition, some evidence has shown that estradiol has a significant effect on atrial electrophysiology, which may be related to the prevention of AF (29). This may also be one reason why women have a lower incidence of AF than men. Citrulline, which is a natural precursor of mammalian arginine biosynthesis, is significantly upregulated after catheter ablation in AF patients (30). Notably, supplementation with citrulline can increase the circulating level of arginine more effectively than direct supplementation with arginine and nitric oxide (NO), which plays an important role in maintaining the integrity and stability of vascular endothelial cells and regulating blood flow and organ circulation. This effect is catalyzed by eNOS and relies heavily on the availability of arginine (31, 32). Postoperatively upregulated 6-hydroxymelatonin has also been shown to eliminate the effects of free radical inhibition of lipid peroxidation and protect against neuronal damage caused by the neurotoxin $Fe^{2+}$ (33).

We found that some decreased metabolites in the AF patients, such as caffeine, ascorbic acid, eicosapentaenoic acid, and oleanolic acid, were closely related to diet, and most could be derived from a plant-based diet. Available evidence suggests that moderate caffeine intake (400 to 600 mg/day) is not associated with overall cardiovascular disease, arrhythmias, heart failure, blood pressure changes, or an increased risk of HTN in healthy people. In contrast, moderate caffeine intake is associated with a lower risk of cardiovascular disease, mortality, and all-cause mortality (34, 35). One study

even found that higher caffeine intake ($>$320 mg/day) was associated with a lower incidence of AF (36). Evidence shows that perioperative ascorbic acid therapy in patients undergoing cardiac surgery can reduce the frequency of AF after cardiac surgery and shorten the length of stay in the intensive care unit (ICU) and total inpatient time (37). This may be related to the fact that ascorbic acid can reduce inflammation and oxidative stress before the occurrence of AF and reduce the formation of peroxynitrite and electrical remodeling induced by atrial pacemaking (38). Eicosapentaenoic acid (EPA) has a wide range of cardiovascular protective effects, including prevention of arrhythmias, improvement of ventricular diastolic anti-inflammatory response, enhancement of plaque stability, and antiatherosclerosis effects (39, 40). Oleanolic acid is a pentacyclic triterpenoid compound (41) that can protect the myocardium from ischemic injury and has antioxidant, antihyperlipidemic, antiarrhythmic, and membrane stabilization effects. Inhibition of the Akt/mammalian target of rapamycin (mTOR) signaling pathway may play a very important role in alleviating the development of heart remodeling caused by stress overload (42–45).

Meanwhile, our results show that caffeine was negatively correlated with *Klebsiella*, and estradiol and ascorbic acid were positively correlated with *Agathobacter*. In AF patients, *Klebsiella* increased, caffeine was downregulated, *Agathobacter* decreased, and estradiol and ascorbate were downregulated. Therefore, the remarkable alterations of these metabolites in AF patients, which were influenced by intake or gut microbiota, reduce the protective effects on cardiac function and may increase the susceptibility to AF and aggravate its progression.

In addition, as an important complication of AF, ischemic stroke has an approximately 20% mortality rate and 60% disability rate. The gut-heart and brain-gut axes may provide further directions for the prevention and treatment of AF-related stroke. Yin et al. (17) found that intestinal symbiotic bacteria or beneficial bacteria were decreased, while opportunistic pathogenic bacteria were increased in patients with ischemic stroke, and remodeling the gut microbiota through enrichment with probiotics and supplementation with butyric acid may be an effective method for treating ischemic stroke (46). In our study, in the short term after catheter ablation, intestinal symbiotic bacteria and beneficial bacteria (including some bacteria that produce butyric acid) were increased in AF patients, opportunistic pathogenic bacteria were decreased, and some metabolites, such as citrulline, that were increased after surgery had indirect regulatory effects on endothelial cell function. Hence, catheter ablation may reduce the risk of AF-related stroke by changing the gut microbiota and metabolites of AF patients. Certainly, longer follow-up and more cohort studies should be conducted to prove this hypothesis, which is the focus of our follow-up work.

There are some limitations of our study. First, the number of patients was small, and a longer, large-scale study is needed. Second, although numerous studies have shown that HTN and type 2 diabetes mellitus (T2DM) are associated with gut microbiota and metabolites, we found that most patients with AF had HTN. To reflect the authenticity of the clinic, we retained subjects with HTN and T2DM in the AF group and matched for HTN and T2DM in the control group. We intended to collect blood samples from the subjects for a metabolomics study, but, considering that this is an invasive procedure with multiple collections required during the follow-up period, most patients refused; therefore, our study carried out only metabolomics analysis of feces.

In conclusion, our results showed changes in the gut microbiota and metabolites in AF patients, providing new clues for the underlying pathophysiological mechanisms of AF. We also expounded on the patterns of changes in the gut microbiota and metabolome in AF patients in the short term after catheter ablation, and we preliminarily discussed the effects of diet on AF from the perspective of gut metabolomics, laying the foundation for later studies.

## MATERIALS AND METHODS

**Study cohort.** Individuals with a history of heart failure, coronary heart disease, structural heart disease, comorbidities (inflammatory bowel diseases, irritable bowel syndrome, autoimmune diseases, liver diseases, renal diseases, or cancer) or the use of antibiotics or probiotics in the past month were excluded. We

recruited 50 patients with nonvalvular AF who were hospitalized in Nanfang Hospital of Southern Medical University from December 2019 to September 2020, and of these, 36 AF patients were eventually included in our study (5 patients withdrew from the study, 6 patients did not provide fecal samples, and 3 patients provided fecal samples that were not eligible). At the same time, 30 control subjects matched with the experimental group for sex, age, BMI, hypertension, and diabetes history were recruited from the Health Management Center of Southern Medical University, and fecal samples were collected. We then selected 30 of the 36 patients who received catheter ablation treatment, and ultimately, 18 patients completed fecal collection three times (before surgery, 2 days after surgery, and 1 month after surgery). Fecal samples collected from 8 AF patients 2 days before and after catheter ablation and 10 normal controls were used to study metabolites. The study conformed to the principles of the Declaration of Helsinki. The research protocol was approved by the ethics committee of Nanfang Hospital. All of the participants signed informed consent.

**Sample collection and bacterial DNA extraction.** Stool samples for intestinal microflora detection were collected using IGE Biotechnology Co., Ltd. (Guangzhou, China), fecal collectors, immediately frozen at $-20°C$, transported on ice to the laboratory, and then stored at $-80°C$. Samples used for metabolite detection were collected with a disposable feces collector, immediately transferred into 2-mL cryopreservation tubes, frozen with liquid nitrogen for 15 min, and then stored in a $-80°C$ freezer. Bacterial DNA was extracted using a Tiangen kit (DP712, Tiangen Biotech Co., Ltd., Beijing, China) at Novogene Bioinformatics Technology Co., Ltd. (Beijing, China).

**Amplification and sequencing of the 16S rRNA gene.** Bacterial DNA in fecal samples was amplified by PCR. The following primers were used to amplify the V3-V4 region genes of bacterial 16S rRNA: forward, 5′-TCGTCGGCAGCGTCAGATGTGTATAAGCGACAGCCTACGGGNGNGGCWGCAG-3′, and reverse, 5′-GTCTCGTGGGCTCGGAGATGTGTATAAGAGACAGGACTACHVGGGTATCTAATCC-3′. The amplicon was sequenced on an Illumina NovaSeq sequencing platform at Novogene Bioinformatics Technology Co., Ltd. (Beijing, China). All valid data were clustered by Uparse software (V7.0.1001). Sequences were clustered into OTUs according to 97% similarity, and the sequence with the highest frequency among the OTUs was selected as the representative sequence according to its algorithm principle. Then, species annotation analysis was carried out using the Mothur method and the SSUrRNA database of SILVA138 (set at a threshold of 0.8 to 1). Taxonomic information was obtained, and the species composition of each sample was quantified at each taxonomic level.

**Enterotype analysis and function prediction.** First, the JSD distance matrix was calculated based on species abundance at the genus level. Then, according to the results, the optimal cluster number was selected according to the Calinski-Harabasz (CH) index. Finally, the partitioning around medoids (PAM) clustering algorithm was used for clustering analysis. Samples clustered into the same group belonged to the same gut enterotype. PCoA dimension reduction was conducted according to the obtained clustering groups. At the same time, based on the species abundance information and the results of enterotype analysis, species with significant differences among enterotypes were identified by statistical testing. Species with significant differences and the highest relative abundance among enterotypes were defined as the name of the enterotype.

**Metabolomic analysis based on LC-MS.** Fecal samples (100 mg) were pipetted into microcentrifuge tubes (1.5-mL), and an 80% methanol aqueous solution (500 $\mu$L) was added. Each tube was placed on an oscillator, oscillated until the mixture was evenly mixed, and then incubated in an ice bath for 5 min. Then, the samples were centrifuged at 15,000 rpm for 20 min at 4°C. The supernatant was diluted with mass spectrometry-grade water to a methanol content of 53%. After centrifugation, the supernatant was collected and analyzed by LC-MS. Equal volumes of supernatant from each test sample were mixed as a quality control sample. The blank sample was 53% methanol aqueous solution, and the pretreatment process was the same as that of the experimental sample.

**Data availability.** Data were deposited at NCBI under accession number SUB10756353.

## SUPPLEMENTAL MATERIAL

Supplemental material is available online only.
**SUPPLEMENTAL FILE 1**, PDF file, 6.1 MB.

## ACKNOWLEDGMENTS

This work was funded by President Foundation of Nanfang Hospital, Southern Medical University (2019Z002).

Conceptualization: Y.W., Y.B. Data curation: K.H. and Q.L. Methodology: K.H. Project administration: K.H. Resources: Y.W., Y.B., Q.Y., and S.W. Validation: Y.B. Formal analysis: Q.L. Investigation: X.L. Funding acquisition: Y.W. Software: Q.L. and X.L. Visualization: X.L. and H.X. Writing-original draft: K.H. Writing-review and editing: K.H., Y.W., and Y.B. Supervision: Y.W. and Q.Y.

We declare no conflicts of interest.

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
