## [Reviewer comments · Microbiology Spectrum]

Microbiology Spectrum

Gut microbiota and metabolites in atrial fibrillation patients and their changes after catheter ablation

Kang Huang, Yuegang Wang, Yang Bai, Qiuyan Luo, Xuancai Lin, Qiuyu Yang, Shihao Wang, and Hongjie Xin

Corresponding Author(s): Yuegang Wang, Nanfang Hospital

Review Timeline:

Submission Date:	July 29, 2021
Editorial Decision:	August 26, 2021
Revision Received:	October 26, 2021
Editorial Decision:	November 8, 2021
Revision Received:	January 6, 2022
Editorial Decision:	January 19, 2022
Revision Received:	February 11, 2022
Accepted:	February 28, 2022

Editor: Yongjun Sui

Reviewer(s): Disclosure of reviewer identity is with reference to reviewer comments included in decision letter(s). The following individuals involved in review of your submission have agreed to reveal their identity: Bo Cui (Reviewer #2)

Transaction Report:

DOI: <https://doi.org/10.1128/Spectrum.01077-21>

August 26, 2021

Prof. Yuegang Wang
Nanfang Hospital
Guangzhou Avenue North 1838
Guangzhou City, Guangdong Province
China

Re: Spectrum01077-21 (**Gut microbiota and metabolites in atrial fibrillation patients and their changes after catheter ablation**)

Dear Prof. Yuegang Wang:

Thank you for submitting your manuscript to Microbiology Spectrum. When submitting the revised version of your paper, please provide (1) point-by-point responses to the issues raised by the reviewers as file type "Response to Reviewers," not in your cover letter, and (2) a PDF file that indicates the changes from the original submission (by highlighting or underlining the changes) as file type "Marked Up Manuscript - For Review Only". Please use this link to submit your revised manuscript - we strongly recommend that you submit your paper within the next 60 days or reach out to me. Detailed information on submitting your revised paper are below.

Link Not Available

Sincerely,

Yongjun Sui

Journals Department
Reviewer comments:

Reviewer #1 (Comments for the Author):

The authors performed 16S amplicon sequencing of 30 fecal samples of patients with atrial fibrillation and 30 controls. From this set, 18 patients provided longitudinal samples before and after catheter ablation. Then, only 8 atrial fibrillation patients and 10 controls were included for the metabolomics analysis. I list here some major and minor issues:

The principal issues:

- 1.) Figures need a lot of changes. The labels are wrong (specified in the minor changes section). The panels are not ordered according to the manuscript text. The font size is often so small that it is not readable. The text and figures is shrink, which is not nice at all. The patients groups should have consistent colors in all panels in all figures.
- 2.) I disagree with using the KEGG predictions based on the 16S amplicon data. This approach has been criticized among scientists. It's useful when you compare very different samples, such as gut and soil, but it's very inaccurate when you are trying to compare samples of patients and controls. This approach does not add anything good to this manuscript. The authors have done the metabolomics analysis which is much better than the computational predictions. Importantly, the results of this

analysis do not correspond with the metabolomics results. I recommend removing it from the manuscript.

3.) Discussion is very long and is more about what has been found in other studies than about the present study. Many of the information could be moved to the introduction. For the discussion, cite only those papers which are directly connected with the results of this study. Explain the take home message more clearly. I like that the authors include metabolomics in their study, but the results of metabolomics seem disconnected from the microbiome composition in the discussion section.

4.) I would like to see a Figure containing barplots for all individuals in this study. The barplots in this manuscript are just an average of all individuals, but there could be individuals who differ from the majority in their group. This could be a Supplementary Figure. In addition, I would like to know whether the longitudinal samples of each individual have the same pattern.

Other issues:

1.) Underline in the abstract the biggest strength of the study which is that it includes longitudinal series or samples.

2.) Introduction: What was the reason why the previous studies produced different results? What is the rationale for this study? How does it differ from the previous studies?

3.) Lines 110-112 and 182-183. Specify which are the characteristic species for each enterotype?

4.) Is it possible to label the dots in the PCA plots by patients ID?

5.) The paragraph lines 250-265 contains only numbers and is very disconnected from the following paragraph. It would be nicer to have the numbers together with the concrete names of the metabolites from the following paragraph.

Minor errors:

1.) Line 22: metabonomics should be metabolomic

2.) Do not copy-paste names of bacteria directly from the bioinformatic programs into the abstract, such as *Ruminococcus_gnavus_group*. These computer generated names are useless for comparison with other studies and in the future these names will be updated due to changing taxonomic nomenclature.

3.) Explain the abbreviations, e.g. line 49 ECG, line 251: VIP and FC

4.) Mention only the most important bacteria, because the proportion of bacteria is obvious from the Figure 2

5.) Line 163: I guess there should be AF2 group instead of AF1

6.) Line 211: please, double check that the Figure 1 here is correct

7.) Line 208: please, double check that the panel Figure 4G is correct

8.) Line 223: I don't see any panel 4H (anyway, I recommend to remove the whole paragraph about the KEGG predictions)

9.) Line 231: I don't see any panel 4I (anyway, I recommend to remove the whole paragraph about the KEGG predictions)

Reviewer #2 (Comments for the Author):

This study investigated the effects of atrial fibrillation and catheter ablation on intestinal flora and metabolomics, and defined the changes of intestinal flora and metabolites in patients with AF. Although the authors have drawn rich results, the relevant conclusions can not be fully supported, such as the pathophysiological mechanism of AF and the effect of diet on AF. The relevant results and conclusions need to be further analyzed and summarized.

1. Whether atrial fibrillation and catheter ablation have a direct or indirect impact on intestinal flora and metabolomics, and what mechanisms may play a role?

2. How to rule out the effects of anticoagulant or PPI on intestinal flora and metabolomics?

3. The authors suggest that "this study will help to explore the pathophysiological mechanism of atrial fibrillation and provide an important basis for the early diagnosis, treatment and treatment of complications of atrial fibrillation", however, how to clarify the relationship between "atrial fibrillation and catheter ablation" and "intestinal flora and its metabolomics changes"? Is the former causing the exception of the latter, or the latter causing the occurrence of the former? The author analyzed the correlation between intestinal flora and metabolome changes. The causal relationship between them is relatively clear, and it is not the focus of this paper, so the correlation analysis is of little significance. It is suggested that the correlation analysis between the clinical physiological and biochemical indexes of patients and the changes of intestinal flora and its metabolomics may be of greater significance for causal analysis.

4. The overall expression of this paper is not concise enough, especially the expression of the results and discussion is too long and the focus is not prominent enough; Some English expressions are not standardized and need to be further polished, such as "we found that there were 2692 OTUs in the three groups combined, for a total of 1026 overall; The AF1 group had 280

unique OTUs, the af2 group had 503 unique OTUs, and the af3 group had 199 OTUs ", the meaning of which cannot be clearly understood by the reviewer.

5. The statement of group division in the manuscript is not clear, especially that the statements before and after the grouping basis of AF1, af2 and af3 groups are inconsistent (lines 162-164 vs. lines 240-243), and the reviewer cannot fully understand its meaning.

6. What does lines 761-766 mean?

7. Figure 1-5: lack of sample size data; Some fonts are too small to be seen clearly; The alignment of lines in Table 1 and S2 is not standard, and the intelligibility of the table is poor.

8. The content of the abstract is not complete and lacks a clear concluding statement.

Staff Comments:

Preparing Revision Guidelines

Please return the manuscript within 60 days; if you cannot complete the modification within this time period, please contact me. If you do not wish to modify the manuscript and prefer to submit it to another journal, please notify me of your decision immediately so that the manuscript may be formally withdrawn from consideration by Microbiology Spectrum.

If you would like to submit an image for consideration as the Featured Image for an issue, please contact Spectrum staff.

Dear editor,

Thank you for allowing us to submit a revised draft of the manuscript “Gut microbiota and metabolites in atrial fibrillation patients and their changes after catheter ablation” for publication in the *Microbiology Spectrum*. We appreciate the time and effort that you and the reviewers dedicated to providing feedback on our manuscript and are grateful for the insightful comments on and valuable improvements to our paper. We have substantially revised our manuscript after reading the comments provided by the two reviewers. We have incorporated most of the suggestions made by the reviewers. Those changes are highlighted in red within the manuscript.

Answers to reviewers:

Reviewer #1 (Comments for the Author):

Thank you for the insightful critiques and suggestions, we have revised the manuscript carefully. Here we respond to your comments point by point.

The authors performed 16S amplicon sequencing of 30 fecal samples of patients with atrial fibrillation and 30 controls. From this set, 18 patients provided longitudinal samples before and after catheter ablation. Then, only 8 atrial fibrillation patients and 10 controls were included for the metabolomics analysis.

The principal issues:

1) Figures need a lot of changes. The labels are wrong (specified in the minor changes section). The panels are not ordered according to the manuscript text. The font size is often so small that it is not readable. The text and figures is shrink, which is not nice at all. The patients groups should have consistent colors in all panels in all figures.

Response: We are very sorry for our mistakes, and we have resized the text and figures, we have also revised the colors and labels, Please refer to all the latest uploaded figures.

2) I disagree with using the KEGG predictions based on the 16S amplicon data. This approach has been criticized among scientists. It's useful when you compare very different samples, such as gut and soil, but it's very inaccurate when you are trying to compare samples of patients and controls. This approach does not add anything good to this manuscript. The authors have done the metabolomics analysis which is much better than the computational predictions. Importantly, the results of this analysis do not correspond with the metabolomics results. I recommend removing it from the manuscript.

Response: Thank you very much for your recommendation, we are deeply aware that this approach is not applicable in this study, so we remove this part from the manuscript, please refer to the latest uploaded manuscript.

3) Discussion is very long and is more about what has been found in other studies than about the present study. Many of the information could be moved to the introduction. For the discussion, cite only those papers which are directly connected with the results of this study. Explain the take home message more clearly. I like that the authors include metabolomics in their study, but the results of metabolomics seem disconnected from the microbiome composition in the discussion section.

Response: Thank you very much for this comment, we have streamlined the discussion section, and we also increase the association between gut microbiota and metabolites in the discussion section(L392-398), please refer to the latest uploaded manuscript.

4.) I would like to see a Figure containing barplots for all individuals in this study. The barplots in this manuscript are just an average of all individuals, but there could be individuals who differ from the majority in their group. This could be a Supplementary Figure. In addition, I would like to know whether the longitudinal samples of each individual have the same pattern.

Response: Thank you for this comment. We have uploaded a Supplementary Figure containing barplots for all individuals in this study, please refer to Supplementary Figure A,B,C,D. In addition, when we were conducting a longitudinal study, we calculated the number or rate of individuals who changed postoperative, such as Shannon index and Chao1 index(Please refer to the latest uploaded figure3H, I), and differential gut microbiota(Please refer to the latest uploaded Supplementary Figure F, G).

Other issues:

1.) Underline in the abstract the biggest strength of the study which is that it includes longitudinal series or samples.

Response: Thank you very much for this advice, we have added it in the abstract, please refer to the latest uploaded manuscript(L34-36).

2.) Introduction: What was the reason why the previous studies produced different results? What is the rationale for this study? How does it differ from the previous studies?

Response: In our consideration, the differences in dietary habits and differences between inclusion and exclusion criteria are responsible for the different results. In our study, we tried our best to match confounders such as age, sex, BMI and so on, in addition, the analysis of gut metabolites also reflects dietary habits to a certain extent. Different from previous studies, we conducted horizontal and longitudinal analyses of the gut microbiota and metabolites of AF patients. We have added above content in the abstract, please refer to the latest uploaded manuscript.

3.) Lines 110-112 and 182-183. Specify which are the characteristic species for each enterotype?

Response: Thank you for this advice. We have specified the characteristic species for each enterotype. please refer to the latest uploaded manuscript(lines 135-136 and 207-209 in red).

4.) Is it possible to label the dots in the PCA plots by patients ID?

Response: Thank you for this advice. When we label the dots in the PCA plots by patients ID, the PCA plots are not clear, so we chose to remove patients ID finally.

5.) The paragraph lines 250-265 contains only numbers and is very disconnected from the following paragraph. It would be nicer to have the numbers together with the concrete names of the metabolites from the following paragraph.

Response: We agree with the reviewer' s comment. When we add the concrete names of the metabolites, it seems repetitive compared with the following paragraph, so we streamline this part finally, please refer to the latest uploaded manuscript(lines 260-264 in red).

Minor errors:

1.) Line 22: metabonomics should be metabolomic

Response: We have revised this, please refer to the latest uploaded manuscript(line 22 in red).

2.) Do not copy-paste names of bacteria directly from the bioinformatic programs into the abstract, such as *Ruminococcus_gnavus_group*. These computer generated names are useless for

comparison with other studies and in the future these names will be updated due to changing taxonomic nomenclature.

Response: We are very sorry for our mistake, and we have revised this in the abstract, please refer to the latest uploaded manuscript.

3.) Explain the abbreviations, e.g. line 49 ECG, line 251: VIP and FC

Response: We have explained the abbreviations, ECG means electrocardiography, VIP means variable importance in the projection, FC means fold change, please refer to the latest uploaded manuscript(line 60 and lines 260-261 in red).

4.) Mention only the most important bacteria, because the proportion of bacteria is obvious from the Figure 2

Response: Thank you for your advice. We list the proportion of bacteria in the top 5 to make the results even clearer, so we keep it finally.

5.) Line 163: I guess there should be AF2 group instead of AF1

Response: We have revised this mistake, please refer to the latest uploaded manuscript(line 191 in red).

6.) Line 211: please, double check that the Figure 1 here is correct

Response: We have revised this mistake, please refer to the latest uploaded manuscript(line 243 in red).

7.) Line 208: please, double check that the panel Figure 4G is correct

Response: We have revised this mistake, please refer to the latest uploaded manuscript(line 235 in red).

8.) Line 223: I don't see any panel 4H (anyway, I recommend to remove the whole paragraph about the KEGG predictions)

Response: Thank you very much for your recommendation. We have revised this mistake and removed the whole paragraph about the KEGG predictions, please refer to the latest uploaded manuscript.

9.) Line 231: I don't see any panel 4I (anyway, I recommend to remove the whole paragraph about the KEGG predictions)

Response: Thank you very much for your recommendation. We have revised this mistake and removed the whole paragraph about the KEGG predictions, please refer to the latest uploaded manuscript.

We have responded all the concerns point by point proposed by Reviewer #1, and have revised the manuscript substantially. Therefore, related data, figures, text and supplementary materials have been adjusted correspondingly and the changed parts in the text are indicated by red fonts. We hope that our manuscript has greatly improved and will satisfy you. Thank you again for your thorough review and precious comments.

Reviewer #2 (Comments for the Author):

Thank you for the insightful critiques and suggestions, we have revised the manuscript carefully. Here we respond to your comments point by point.

This study investigated the effects of atrial fibrillation and catheter ablation on intestinal flora and metabolomics, and defined the changes of intestinal flora and metabolites in patients with AF. Although the authors have drawn rich results, the relevant conclusions can not be fully supported, such as the pathophysiological mechanism of AF and the effect of diet on AF. The relevant results and conclusions need to be further analyzed and summarized.

1. Whether atrial fibrillation and catheter ablation have a direct or indirect impact on intestinal flora and metabolomics, and what mechanisms may play a role?

Response: Thank you very much for your question. To our knowledge, there are very few mechanistic studies on the gut microbiota and atrial fibrillation, a recent article suggests that gut microbiota dysbiosis promotes age-related atrial fibrillation by lipopolysaccharide and glucose-induced activation of NLRP3-inflammasome (PMID: 33757127). Our study also found that *Klebsiella*, *Haemophilus* were significantly increased in patients with AF, and the overgrowth of gram-negative pathogenic bacteria may promote the development of AF through the LPS-inflammatory response system (L318-320). In addition, many gut metabolites can affect heart function, such as FAD (PMID:32540485), caffeine (PMID:29692210), ascorbic acid (PMID : 27806938) and so on, which was also mentioned in our study. Catheter ablation may regulate the gut microbiota of AF, resulting in changes in multiple gut metabolites, which has an indirect regulatory effect on intestinal function stabilization of AF.

2. How to rule out the effects of anticoagulant or PPI on intestinal flora and metabolomics?

Response: Thank you very much for your question. To rule out the effects of anticoagulant or PPI, we grouped AF patients by whether PPIs or anticoagulant were used, and found that most of the differential gut microbiota were not affected significantly by anticoagulant or PPI except *Streptococcus* and *Weissella* (L333-341), please refer to the latest uploaded manuscript.

3. The authors suggest that "this study will help to explore the pathophysiological mechanism of atrial fibrillation and provide an important basis for the early diagnosis, treatment and treatment of complications of atrial fibrillation", however, how to clarify the relationship between "atrial fibrillation and catheter ablation" and "intestinal flora and its metabolomics changes"? Is the former causing the exception of the latter, or the latter causing the occurrence of the former? The author analyzed the correlation between intestinal flora and metabolome changes. The causal relationship between them is relatively clear, and it is not the focus of this paper, so the correlation analysis is of little significance. It is suggested that the correlation analysis between the clinical physiological and biochemical indexes of patients and the changes of intestinal flora and its metabolomics may be of greater significance for causal analysis.

Response: Thank you very much for this comment. To our knowledge, the causal relationship between "atrial fibrillation and catheter ablation" and "intestinal flora and its metabolomics changes" is uncertain by far. More evidence indicates that they are causal to each other, for example, cardiovascular disease can cause the change of gut microbiota and metabolites, but the

change of gut microbiota and metabolites can also promote the development of cardiovascular disease. So we need more evidence to prove it, and this is why we analyzed the correlation between intestinal flora and metabolome changes. In addition, We really agree with the correlation analysis between the clinical physiological and biochemical indexes of patients and the changes of gut microbiota and metabolites, but in our study, there was no significant difference between patients with AF and controls in terms of most clinical physiological and biochemical indexes, so we removed the correlation analysis, we would investigate this aspect in our future studies.

4. The overall expression of this paper is not concise enough, especially the expression of the results and discussion is too long and the focus is not prominent enough; Some English expressions are not standardized and need to be further polished, such as "we found that there were 2692 OTUs in the three groups combined, for a total of 1026 overall; The AF1 group had 280 unique OTUs, the af2 group had 503 unique OTUs, and the af3 group had 199 OTUs ", the meaning of which cannot be clearly understood by the reviewer.

Response: We are sorry to confuse you because of our mistake. We have streamlined the manuscript, and we have modified the unstandard English expression, please refer to the latest uploaded manuscript(lines 192-194 in red).

5. The statement of group division in the manuscript is not clear, especially that the statements before and after the grouping basis of AF1, af2 and af3 groups are inconsistent (lines 162-164 vs. lines 240-243), and the reviewer cannot fully understand its meaning.

Response: We are sorry to confuse you because of our mistake. We have modified the statement of group division. When the analysis of gut microbiota was performed, patients with atrial fibrillation prior to catheter ablation were tagged as group AF1, patients with atrial fibrillation at 2 days after catheter ablation were tagged as group AF2, and patients with atrial fibrillation at 1 month after catheter ablation were tagged as group AF3. Please refer to the latest uploaded manuscript(lines 189-192 in red). When the analysis of gut metabolites was performed, patients with atrial fibrillation prior to catheter ablation were tagged as group AF1, patients with atrial fibrillation at 2 days after catheter ablation were tagged as group AF2, the CG group represented control subjects(lines 249-252 in red).

6. What does lines 761-766 mean?

Response: We are sorry to confuse you because of our mistake. This is the proofreading report of references, we have deleted it.

7. Figure 1-5: lack of sample size data; Some fonts are too small to be seen clearly; The alignment of lines in Table 1 and S2 is not standard, and the intelligibility of the table is poor.

Response: Thank you very much for this comment. We have resized the tables and figures, and the sample size is shown in the material and methods, we have also revised the colors and labels, please refer to all the latest uploaded figures and tables.

8. The content of the abstract is not complete and lacks a clear concluding statement.

Response: Thank you very much for this comment. We have added a concluding statement in the abstract(L34-36), please refer to the latest uploaded manuscript.

We have responded all the concerns point by point proposed by Reviewer #1, and have revised the manuscript substantially. Therefore, related data, figures, text and supplementary materials have been adjusted correspondingly and the changed parts in the text are indicated by red fonts. We hope that our manuscript has greatly improved and will satisfy you. Thank you again for your thorough review and precious comments.

Special thanks to you for your good comments. we tried our best to improve the manuscript and made some changes to the manuscript. we appreciate Editors/Reviewers' warm work earnestly, and hope the correction will meet with approval. Once again, thank you very much for your comments and suggestions.

Sincerely,

Prof. Yuegang Wang

Department of Cardiology, State Key Laboratory of Organ Failure Research, Nanfang Hospital, Southern Medical University, Guangzhou, 510515, China.

E-mail address: 1248508@qq.com.

November 8, 2021

Prof. Yuegang Wang
Nanfang Hospital
Guangzhou Avenue North 1838
Guangzhou City, Guangdong Province
China

Re: Spectrum01077-21R1 (**Gut microbiota and metabolites in atrial fibrillation patients and their changes after catheter ablation**)

Dear Prof. Yuegang Wang:

Thank you for submitting your manuscript to Microbiology Spectrum. When submitting the revised version of your paper, please provide (1) point-by-point responses to the issues raised by the reviewers as file type "Response to Reviewers," not in your cover letter, and (2) a PDF file that indicates the changes from the original submission (by highlighting or underlining the changes) as file type "Marked Up Manuscript - For Review Only". Please use this link to submit your revised manuscript - we strongly recommend that you submit your paper within the next 60 days or reach out to me. Detailed information on submitting your revised paper are below.

Link Not Available

Sincerely,

Yongjun Sui

Journals Department
Reviewer comments:

Reviewer #1 (Comments for the Author):

I'm very disappointed by the changes performed by the authors after my first review. I asked them to improve their figures - I asked them to have common color code for enterotypes and sample groups, to use the same font type, to avoid shrinking the text... Very few has been changed in these figures. The figures are very confusing, they names of the bacteria have been shortened, so it's not comprehensible. Basically, my feeling is that the authors just copy-pasted figures produced by different programs, shrink them so they fit into the maximum figure size and don't really care whether the readers can get the take home message from these figures.

I asked the authors to provide barplots of the bacterial composition for each sample. They created some new figures, but again, they are incomprehensible - only few sample names are visible. The colors for the bacteria are very confusing. I cannot evaluate the manuscript properly if I don't see these data.

I asked the authors to remove the KEGG functional prediction - they have remove something, but forget to remove the KEGG info from the paragraph lines 490-499.

The abstract is just a list of species which have been found in different proportions among the groups, but there is no connection of the bacteria to the metabolites. Make sure the readers get the take-home message from the abstract. It needs to be rewritten

Sequences need to be deposited to public repositories. Add the sample accession numbers (or study ID) to the manuscript, so the data will be publicly accessible.

Reviewer #2 (Comments for the Author):

The content of the manuscript has been well revised in accordance with the concerns of the reviewers.

Staff Comments:

Preparing Revision Guidelines

Please return the manuscript within 60 days; if you cannot complete the modification within this time period, please contact me. If you do not wish to modify the manuscript and prefer to submit it to another journal, please notify me of your decision immediately so that the manuscript may be formally withdrawn from consideration by Microbiology Spectrum.

Dear editor,

Thank you for allowing us to submit a revised draft of the manuscript “Gut microbiota and metabolites in atrial fibrillation patients and their changes after catheter ablation” for publication in the Microbiology Spectrum. We appreciate the time and effort that you and the reviewers dedicated to providing feedback on our manuscript and are grateful for the insightful comments on and valuable improvements to our paper. We have substantially revised our manuscript after reading the comments provided by the two reviewers. We have incorporated the suggestions made by the reviewers. Those changes are highlighted in red within the manuscript.

Answers to reviewers:

Reviewer #1 (Comments for the Author):

Thank you for the insightful critiques and suggestions, we have revised the manuscript carefully. Here we respond to your comments point by point.

1) I asked them to have common color code for enterotypes and sample groups, to use the same font type, to avoid shrinking the text.

Response: We sincerely apologize for our mistake, and we have carefully modified each image to ensure common color code for enterotypes and sample groups, and we have used the same font type. Please refer to all the latest uploaded figures.

2) I asked the authors to provide barplots of the bacterial composition for each sample. They created some new figures, but again, they are incomprehensible - only few sample names are visible. The colors for the bacteria are very confusing.

Response: We sincerely apologize for our mistake, and we have re-modified and uploaded the figures - all sample names are visible, and we have re-modified the colors for the bacteria. Please refer to all the latest uploaded Supplementary Figure.

3) I asked the authors to remove the KEGG functional prediction - they have remove something, but forget to remove the KEGG info from the paragraph lines 490-499.

Response: We are very sorry for our mistakes, and we have removed the whole paragraph, please refer to the latest uploaded manuscript.

4) The abstract is just a list of species which have been found in different proportions among the groups, but there is no connection of the bacteria to the metabolites. Make sure the readers get the take-home message from the abstract. It needs to be rewritten.

Response: Thank you very much for your recommendation. We have rewritten the abstract which includes the connection of the bacteria to the metabolites(L18-38), please refer to the latest uploaded manuscript.

5) Sequences need to be deposited to public repositories. Add the sample accession numbers (or study ID) to the manuscript, so the data will be publicly accessible.

Response: Thank you very much for your recommendation. We have uploaded the raw data to the public repositories, the sample accession numbers is SUB10756353(L481-482), please refer to the latest uploaded manuscript.

We have responded all the concerns point by point proposed by Reviewer #1, and have revised the manuscript substantially. Therefore, related data, figures, text and supplementary materials have been adjusted correspondingly and the changed parts in the text are indicated by red fonts. We hope that our manuscript has greatly improved and will satisfy you. Thank you again for your thorough review and precious comments.

Reviewer #2 (Comments for the Author):

The content of the manuscript has been well revised in accordance with the concerns of the reviewers.

Response: Thank you so much for getting your affirmation, and we will continue to work hard.

Special thanks to you for your good comments. we tried our best to improve the manuscript and made some changes to the manuscript. we appreciate Editors/Reviewers' warm work earnestly, and hope the correction will meet with approval. Once again, thank you very much for your comments and suggestions.

Sincerely,

Prof. Yuegang Wang

Department of Cardiology, State Key Laboratory of Organ Failure Research, Nanfang Hospital, Southern Medical University, Guangzhou, 510515, China.

E-mail address: 1248508@qq.com.

January 19, 2022

Prof. Yuegang Wang
Nanfang Hospital
Guangzhou Avenue North 1838
Guangzhou City, Guangdong Province
China

Re: Spectrum01077-21R2 (**Gut microbiota and metabolites in atrial fibrillation patients and their changes after catheter ablation**)

Dear Prof. Yuegang Wang:

Link Not Available

Sincerely,

Yongjun Sui

Journals Department
Reviewer comments:

Reviewer #1 (Comments for the Author):

I thank the authors for changing the figures, so now they are finally readable. I finally could compare the information from the main text with the information shown in the figures. This was not possible in the previous versions.

Unfortunately, I still detected a lot of issues:

1) It looks like the panels C and D of the Figure 6 are switched. The figure legend says that the panel C shows analysis between the AF and control groups and the panel D shows differences before and after AF catheter ablation. I detected that it does not match the abstract (lines 31-33) and the main text lines 286-287. Abstract describes a positive correlation of citrulline and with *Ralstonia* and *Lactobacillus* and a negative correlation of oleanolic acid with *Ralstonia* after catheter ablation, but this is not found in the Figure 6 panel D, which is supposedly showing correlations catheter ablation, but it can be found in the panel C. The same is true for the lines 286-287.

2.) The figure legends is not matching what is shown in the figures. For example, the title of the Figure 3 is "Gut microbiota diversity in AF patients before and after catheter ablation", however, the Figure 3 panel A, B, C, D show data on the comparison between AF and CG which belongs to the Figure 2.

3.) Order of the panels is not matching the order in which they are described in the text. For example, 3A goes before 2D and 2E. Please, revise the order of all panels! The authors should think about a better organization of the panels, across the Figure 2, 3 and 4, because in the current version, the Figure 3 is a hybrid figure showing comparison of AF and CG in the panel A, B, C, D, and the remaining panels show data on the AF1, AF2 and AF3. The Figure 3 panel A, B, C, D, can go to the Figure 2. In the current version the data on the AF1, AF2 and AF3 are distributed between Figure 3 and Figure 4 without any apparent logic.

4.) There are many panels which contain results which are not properly explained in the main text. For example, lines 167-168: "The phylogenetic distribution of the dominant species is shown in the phylogenetic branching diagram (Figure 3A)." The authors did not explain what is the take-home message of this panel 3A. If this panel does not contain any important scientific information, it can be removed. Another example is the supplementary Figure C and D. The main text, lines 239-240 says only: "We also calculated the number or rate of individuals who showed changes postoperatively (Supplementary Figure C, D)." This is another example, where the main text contains only a link to the figure, but does not explain any results, which the reader will see in the figure. The authors must revise the whole manuscript and make sure that the important take-home-messages from all figure panels are properly explained in main text.

5.) Order of the samples in the Supplementary Figure panels G and H does not follow any logical order. I think the samples should be ordered either in a way that we see AF1 AF2 and AF3 from the same individual next to each other, or in a way that we see all samples from AF1 first, followed by all samples in AF2 and AF3, but in that case the samples in these groups must be ordered alphabetically, so we can easily find the AF1 AF2 and AF3 from the same individual.

6.) I asked the authors last time to generate supplementary figures in which we would see bacterial profiles in each patient, because the main text figures show only an average of all patient (e.g. Figures 2C, 2D, 2E, 4I and 4H). Now, after the authors finally updated the figures, I could finally see that the species shown in the Figures 2C, 2D, 2E, 4I and 4H are in such a low proportion, that they are not visible in the Supplementary figures G and H. I am sure that there are many other ways to show the individual differences of proportions of these bacteria. For example, instead of a bars in the Figures 2C, 2D, 2E, 4I and 4H, the authors can use dots. Each dot would represent the exact proportion of the given bacterial species (so called jitter plot). In that case, the species must be divided into two groups which will have their own x-axis: the species which have average proportion higher than 0.01% or lower than 0.01%, otherwise the dots representing abundance lower than 0.01% would be concentrated near 0 and we would not see any differences between groups.

7.) No details on the correlation analysis in the methods. I mean, did the authors use Parson or Spearman correlations?

8.) The text still contains copy-pasted names of bacteria copied directly from the bioinformatic programs into the manuscript, despite I asked the authors to revise it during the first review. I guess that Marseille-P5638, and Marseille-P2931 (line 162) are in fact Dialister and Prevotella.

Staff Comments:

Preparing Revision Guidelines

Please return the manuscript within 60 days; if you cannot complete the modification within this time period, please contact me. If

you do not wish to modify the manuscript and prefer to submit it to another journal, please notify me of your decision immediately so that the manuscript may be formally withdrawn from consideration by Microbiology Spectrum.

Dear editor,

Thank you for allowing us to submit a revised draft of the manuscript “Gut microbiota and metabolites in atrial fibrillation patients and their changes after catheter ablation” for publication in the *Microbiology Spectrum*. We appreciate the time and effort that you and the reviewers dedicated to providing feedback on our manuscript and are grateful for the insightful comments on and valuable improvements to our paper. We have substantially revised our manuscript after reading the comments provided by the two reviewers. We have incorporated the suggestions made by the reviewers. Those changes are highlighted in red within the manuscript. Furthermore, we are sorry that the organization of Kang Huang is changed from Aerospace Center Hospital to Nanfang Hospital, Southern Medical University due to the author's work changes, and we have revised in the manuscript, which are highlighted in red within the title page. Thanks so much for your comprehension.

Answers to reviewers:

Reviewer #1 (Comments for the Author):

I thank the authors for changing the figures, so now they are finally readable. I finally could compare the information from the main text with the information shown in the figures. This was not possible in the previous versions. Unfortunately, I still detected a lot of issues:

1) It looks like the panels C and D of the Figure 6 are switched. The figure legend says that the panel C shows analysis between the AF and control groups and the panel D shows differences before and after AF catheter ablation. I detected that it does not match the abstract (lines 31-33) and the main text lines 286-287. Abstract describes a positive correlation of citrulline and with *Ralstonia* and *Lactobacillus* and a negative correlation of oleanolic acid with *Ralstonia* after catheter ablation, but this is not found in the Figure 6 panel D, which is supposedly showing correlations catheter ablation, but it can be found in the panel C. The same is true for the lines 286-287.

Response: We sincerely apologize for our mistake, the order was transposed when the image was modified, and we have modified the figure and the figure legend. Please refer to all the latest uploaded manuscript(L270-282, L677-680) and figures.

2) The figure legends is not matching what is shown in the figures. For example, the title of the Figure 3 is "Gut microbiota diversity in AF patients before and after catheter ablation", however, the Figure 3 panel A, B, C, D show data on the comparison between AF and CG which belongs to the Figure 2.

Response: Thank you very much for your recommendation. We have modified the distribution of the images to match the figure legends and the figures. Please refer to all the latest uploaded figures(Figure 2, 3).

3) Order of the panels is not matching the order in which they are described in the text. For example, 3A goes before 2D and 2E. Please, revise the order of all panels! The authors should think about a better organization of the panels, across the Figure 2, 3 and 4, because in the current version, the Figure 3 is a hybrid figure showing comparison of AF and CG in the panel A, B, C, D, and the remaining panels show data on the AF1, AF2 and AF3. The Figure 3 panel A, B, C, D, can go to the Figure 2. In the current version the data on the AF1, AF2 and AF3 are distributed between Figure 3 and Figure 4 without any apparent logic.

Response: Thank you very much for your recommendation. We have carefully modified the order of each image and we have readjusted the distribution of the images of the Figure 2, 3 and 4. Please refer to all the latest uploaded figures(Figure 2, 3).

4) There are many panels which contain results which are not properly explained in the main text. For example, lines 167-168: "The phylogenetic distribution of the dominant species is shown in the phylogenetic branching diagram (Figure 3A)." The authors did not explain what is the take-home message of this panel 3A. If this panel does not contain any important scientific information, it can be removed. Another example is the supplementary Figure C and D. The main

text, lines 239-240 says only: "We also calculated the number or rate of individuals who showed changes postoperatively (Supplementary Figure C, D)." This is another example, where the main text contains only a link to the figure, but does not explain any results, which the reader will see in the figure. The authors must revise the whole manuscript and make sure that the important take-home-messages from all figure panels are properly explained in main text.

Response: Thank you very much for your recommendation. Indeed, the phylogenetic branching diagram(Figure 3A) and Supplementary Figure C, D are merely serve as some additional supplementary information, and according to your recommendation, we decide to remove these unimportant information after the discussion. Please refer to all the latest uploaded figures.

5) Order of the samples in the Supplementary Figure panels G and H does not follow any logical order. I think the samples should be ordered either in a way that we see AF1 AF2 and AF3 from the same individual next to each other, or in a way that we see all samples from AF1 first, followed by all samples in AF2 and AF3, but in that case the samples in these groups must be ordered alphabetically, so we can easily find the AF1 AF2 and AF3 from the same individual.

Response: Thank you very much for your recommendation. We have modified the order of the samples alphabetically in the Supplementary Figure panels E and F. Please refer to all the latest uploaded Supplementary Figure.

6) I asked the authors last time to generate supplementary figures in which we would see bacterial profiles in each patient, because the main text figures show only an average of all patient (e.g. Figures 2C, 2D, 2E, 4I and 4H). Now, after the authors finally updated the figures, I could finally see that the species shown in the Figures 2C, 2D, 2E, 4I and 4H are in such a low proportion, that they are not visible in the Supplementary figures G and H. I am sure that there are many other ways to show the individual differences of proportions of these bacteria. For example, instead of a bars in the Figures 2C, 2D, 2E, 4I and 4H, the authors can use dots. Each dot would represent the exact proportion of the given bacterial species (so called jitter plot). In that case, the species must be divided into two groups which will have their own x-axis: the species which have average proportion higher than 0.01% or lower than 0.01%, otherwise the dots representing abundance lower than 0.01% would be concentrated near 0 and we would not see any differences between groups.

Response: Thank you very much for your recommendation. The species have been divided into several groups which have their own x-axis in the Figures 2C, 2D, 2E, 3N and 3O, and the species with low abundance can show differences between groups. We have also tried using dot plots instead of a bars in the Figures 2C, 2D, 2E, 3N and 3O, but we decide to keep the bar graph finally after repeated consideration because it is more clear. Please refer to all the latest uploaded figures.

7.) No details on the correlation analysis in the methods. I mean, did the authors use Pearson or Spearman correlations?

Response: We are very sorry for our mistakes. We use Spearman correlations on the correlation analysis, and we have added it in the manuscript(L682). Please refer to the latest uploaded manuscript.

8.) The text still contains copy-pasted names of bacteria copied directly from the bioinformatic

programs into the manuscript, despite I asked the authors to revise it during the first review. I guess that Marseille-P5638, and Marseille-P2931 (line 162) are in fact Dialister and Prevotella.

Response: Thank you very much for your recommendation. Indeed, Marseille-P5638 belongs to Dialister and Marseille-P2931 belongs to Prevotella, but the species involved in the paragraph are all at the genus level, we decide to remove them after the discussion(L160). Please refer to the latest uploaded manuscript.

We have responded all the concerns point by point proposed by Reviewer #1, and have revised the manuscript substantially. Therefore, related data, figures, text and supplementary materials have been adjusted correspondingly and the changed parts in the text are indicated by red fonts. We hope that our manuscript has greatly improved and will satisfy you. Thank you again for your thorough review and precious comments.

Special thanks to you for your good comments. we tried our best to improve the manuscript and made some changes to the manuscript. we appreciate Editors/Reviewers' warm work earnestly, and hope the correction will meet with approval. Once again, thank you very much for your comments and suggestions.

Sincerely,

Prof. Yuegang Wang

Department of Cardiology, State Key Laboratory of Organ Failure Research, Nanfang Hospital, Southern Medical University, Guangzhou, 510515, China.

E-mail address: 1248508@qq.com.

February 28, 2022

Prof. Yuegang Wang
Nanfang Hospital
Guangzhou Avenue North 1838
Guangzhou City, Guangdong Province
China

Re: Spectrum01077-21R3 (**Gut microbiota and metabolites in atrial fibrillation patients and their changes after catheter ablation**)

Dear Prof. Yuegang Wang:

Your manuscript has been accepted, and I am forwarding it to the ASM Journals Department for publication. You will be notified when your proofs are ready to be viewed.

Sincerely,

Yongjun Sui
Editor, Microbiology Spectrum

Journals Department
Additional file: Accept